# Influence of Chinese Herbal Formula on Bone Characteristics of Cobb Broiler Chickens

**DOI:** 10.3390/genes13101865

**Published:** 2022-10-15

**Authors:** Yong Liu, Shuangmin Liang, Xiannian Zi, Shixiong Yan, Mengqian Liu, Mengyuan Li, Yanhao Zhao, Tengfei Dou, Changrong Ge, Kun Wang, Junjing Jia

**Affiliations:** 1College of Animal Science and Technology, Yunnan Agricultural University, Kunming 650201, China; 2College of Food Science and Technology, Yunnan Agricultural University, Kunming 650201, China

**Keywords:** Cobb broiler, Chinese herbal formula, bisphosphonates, bone metabolism

## Abstract

To evaluate the prevention and treatment effect of a Chinese herbal formula (CHF) on the bone disease of Cobb broiler chickens, compare its efficacy with Bisphosphonates (BPs), and provide a theoretical basis for studying the nutritional regulation technology of CHF to improve the bone characteristics of broiler chickens. In this study, 560 one-day-old Cobb broiler chickens were examined for the influence of Chinese herbal formula (CHF) and Bisphosphonates (BPs). Different doses of CHF and BPs were added to the diet, and the 30- and 60-day-old live weight, tibial bone strength, the microstructure of the distal femur cancellous bone, blood biochemical indexes related to bone metabolism, and genes related to bone metabolism were determined and analyzed. The results showed that the live weight of Cobb broilers fed with CHF and BPs in the diet was as follows: The live weight of the CHF group was higher than that of the normal control (NC) group, while the live weight of the BPs group was lower than that of the NC group; the CHF and BPs improved the bone strength of Cobb broilers and increased the elastic modulus, yield strength, and maximum stress of the tibia. CHF and BPs increased the cancellous bone mineral density (BMD), bone tissue ratio (BV/TV), bone surface area tissue volume ratio (BS/TV), bone trabecular thickness (Tb.Th), and bone trabecular number (Tb.N) in the distal femur, and decreased the bone surface area bone volume ratio (BS/BV) and bone trabecular separation (Tb.Sp). Thus, the microstructure of the bone tissue of the distal femur was improved to a certain extent. Both the CHF and the BPs also increased the serum levels of the vitamin D receptor (VDR), osteoprotegerin (OPG), and alkaline phosphatase (ALP), and decreased the content of osteocalcin (OT). Meanwhile, CHF and BPs upregulated the expression of osteogenic genes (BMP-2, OPG, Runx-2) to promote bone formation and downregulated the expression of osteoclastic genes (RANK, RANKL, TNF-α) to inhibit bone resorption, thus affecting bone metabolism. Conclusion: The CHF could improve the skeletal characteristics of Cobb broilers by upregulating the expression of bone-forming-related genes and downregulating the expression of bone-breaking-related genes, thus preventing and controlling skeletal diseases in Cobb broilers. Its effect was comparable to that of BPs. Meanwhile, the CHF-H group achieved the best results in promoting the growth and improvement of the skeletal characteristics of Cobb broilers based on the live weight and skeletal-characteristics-related indexes.

## 1. Introduction

As an important economic trait of poultry, skeletal characteristics affect the development of the broiler industry. At present, broiler breeds are mainly selected for rapid growth, but rapid weight gain increases the burden on the skeletal system, resulting in abnormal skeletal development, which causes poultry to suffer from bone disease and hinders the sustainable and healthy development of the poultry industry. Therefore, it is of great significance to the poultry industry to improve the bone properties of poultry and to reduce the incidence of bone disease. At present, the treatment of bone diseases mainly relies on chemical drugs, but the use of chemical drugs on poultry can lead to drug residue and certain side effects. Therefore, the search for alternatives with fewer side effects and no effect on human consumption has attracted much attention.

In poultry, osteoporosis is a progressive bone-weakening disease characterized by low bone mineral density and degeneration of the microarchitecture of the bone tissue, resulting in broken bones in poultry. Severe fractures can result in severe morbidity and mortality in poultry. At present, there are two main types of drugs for the treatment of bone diseases: one is bone resorption inhibitors, which mainly inhibit bone resorption; the other is bone formation promoters, which mainly promote bone formation. Bisphosphonates are representative bone resorption inhibitors and are often used in the clinical treatment of osteoporosis. After resorption, they can quickly enter bone tissue and inhibit osteoclast activity and bone resorption [1]. The symptoms described in traditional Chinese medicine under the names of “strain”, “bone paralysis”, “bone atrophy” and “backbone” are similar to the manifestations of modern bone diseases [2]. Chinese herbal formula can be used to treat bone diseases, it has the characteristics of rich drug sources, no drug resistance, no residue, no side effects, no environmental pollution, no drug-induced diseases, easy processing, and low cost. At present, Chinese herbal formula has been reported in the treatment of osteoporosis in humans and other animals. However, there are few reports on the application of the Chinese herbal formula in poultry. In this study, eight kinds of commonly used Chinese herbs for the treatment of bone diseases were selected: Danggui, Lycii Radicis Cortex, Yam, Radix Salviae Miltiorrhizae, Schisandra, Osthole, Yuanhu, and Oyster, all of which can produce certain advantages in the treatment of bone diseases when used in the formation of Chinese herbal formula and can also play a role in the treatment of bone diseases while tonifying Qi and promoting osteoclastogenesis and the expression of osteoprotegerin. Chinese herbal formula has regulatory effects on osteoclast differentiation regulators and certain links in cytokines and hormones closely related to bone metabolism which increases the apoptosis rate of osteoclasts and inhibits their bone resorption by inhibiting RANKL and other related genes, promotes osteogenic differentiation and regulates bone formation through bone morphogenetic proteins and extracellular signal-regulated protein kinase signaling pathways. Rationally applying the Chinese herbal formula to the poultry-rearing process and observing the growth and skeletal characteristics of poultry [3,4,5,6,7,8,9,10] can provide more effective and healthy theoretical guidance for the treatment of bone diseases in poultry.

Therefore, this study compared the addition of CHF and BPs in Cobb broiler feed. From the perspective of broiler bone characteristics, the effect of CHF on the prevention and treatment of broiler bone disease was discussed. The purpose was to develop technical solutions for the prevention and treatment of bone disease in broilers and new drugs or feed additives to reduce the incidence of bone disease in broilers, in order to effectively enhance the bone development of poultry and reduce the incidence of bone disease in broilers. It provided a theoretical basis for the development of pollution-free green feed additives.

## 2. Materials and Methods

### 2.1. Animal Experimentation Ethical Statement

All studies involving animals were conducted in accordance with the Regulations on the Administration of Laboratory Animal Affairs (Ministry of Science and Technology of China; revised June 2004). All procedures conducted with the chickens were approved by the Yunnan Agricultural University Animal Care and Use Committee (approval ID: YAUACU C01). Sample collection was performed in accordance with the “Guide for the Care and Use of Laboratory Animals of Yunnan Agricultural University”.

### 2.2. Chicken, Diet and Housing

This study was carried out at Yunnan Agricultural University (Kunming, China). For the experiment, 560 1-day-old healthy Cobb broilers (Kunming ZhengDa Co., Ltd., Kunming, China) were selected and raised in the teaching practice chicken farm of Yunnan Agricultural University. The experimental diet was formulated according to the nutritional requirements of Chinese broilers (NY/T33-2004) (Table 1) starting diet (Phase I: 20.0% CP and 2900 Kcal/kg ME) to 30 days of age. From 30 days of age onward, chickens were fed a regular diet (Phase II: 19.5% CP and 2860 Kcal/kg ME) for up to 60 days. Immunization was carried out according to routine immunization procedures (Table 2). Both males and females were raised in the same pen with a density of 15 birds/m^2^ from 0 days to 14 days, 6 birds/m^2^ from 15 days to 28 days, and 3 birds/m^2^ from 29 days to 60 days. There was free access to food and water. During the experimental period, the feeding was carried out with reference to the feeding management standard of Yunnan Agricultural University: 1–3 days old, 35–33 °C, relative humidity 65–70%, light intensity 25 Lx, 24 h; 4–7 days old, 33–30 °C, relative humidity 65–70%, light intensity 10 Lx, 23 h; 8–14 days old, 30–28 °C, relative humidity 60–65%, light intensity 10 Lx, 23 h; 15–21 days old, 28–26 °C, relative humidity 55–60%, light intensity 8 Lx, 18 h; 22–28 days old, 26–24 °C, relative humidity 55–60%, light intensity 8 Lx, 18 h; 29–35 days old, 24–21 °C, relative humidity 55% or so, light intensity 8 Lx, 18 h; 36–60 days old, 21–18 °C. The relative humidity is about 55%, the light intensity is 5 Lx, 18 h. Fly and rodent control, cleaning, and disinfection work were regularly carried out.

### 2.3. Extraction of Chinese Herbal Formula

The Chinese herbal formula was composed of Danggui, Lycii Radicis Cortex, Yam, Radix Salviae Miltiorrhizae, Schisandra, Yuanhu, Osthole, and Oyster [11,12], according to 5:5:5:1:5:1.6:2:1.4, with 10, 8, and 6 additions of water to perform 3 ultrasonic extractions at 60 min each time. Then suction filtration was performed, and the filtrate was combined, concentrated into an extract, freeze-dried, and kept for later use.

### 2.4. Feeding Experiment and Slaughter Sampling

560 Cobb broilers (1-day-old) were randomly divided into 7 groups (CHF-H, CHF-M, CHF-L, BPs-H, BPs-M, BPs-L, and NC) with 80 chickens in each group (half male and half female), and CHF extracts and BPs were added to the diet in a certain proportion (Table 3). At 30 and 60 days of age, 30 chickens (half male and half female) were randomly picked from each group for body weight measurements, and they were killed by carotid arterial bleeding. A 5 mL centrifuge tube was used to collect 3 mL blood samples, and they were left standing for 30 min and centrifuged at 3000 rpm for 20 min at room temperature, the serum was stored in a 1.5 mL centrifuge tube and stored in a −20 °C refrigerator for the determination of biochemical indexes related to bone metabolism in the blood. The thoracic vertebrae and leg cartilages were taken in 2 mL cryovials and quickly frozen in liquid nitrogen, and then transferred to a −80 °C ultra-low temperature freezer for preservation for tissue total RNA extraction. The intact tibia and femur of the left leg were collected (Figure 1A), the muscles and fascia were removed, wrapped with gauze dipped in 0.75% normal saline, placed in a ziplock bag, and stored in a −20 °C refrigerator. The tibia was used for the determination of bone strength-related indexes, and the femur was used for the analysis of the microstructure of bone tissue. After the experimental samples were collected, the experimental animal carcasses were disposed of in accordance with the Guide for the Care and Use of Laboratory Animals at Yunnan Agricultural University.

### 2.5. Determination Index and Method

#### 2.5.1. Live Weight Determination

The chickens were stopped from feeding 16 h before slaughter and drinking water at 12 h. The body weight of the chickens was measured at 30 and 60 days of age in the morning. 30 of each (half male and half female) were selected.

#### 2.5.2. Bone Strength Determination

The tibial diaphysis was measured by a three-point bending test using an AG-IS electronic universal testing machine (Shimadzu Corporation, Kyoto, Japan). The diaphyseal length of the intact tibia was measured, the midpoint of the diaphysis was found, and the support gauge was calculated. The intact tibia sample was placed on two support points with a certain support gauge, and a downward load was applied to the tibia sample above the midpoint of the two support points. At this time, the three contact points of the sample formed two equal moments, which meant three-point bending occurred. The loading speed of this process was 2 mm/min, and the loading was uniform until the sample was destroyed. The tibia sample should be kept moist throughout the experiment. The calculation formula of the support rail distance used was: L = (a + 3b) ± 0.5b, where a was the diameter of the support rail, and b was the diameter of the bone sample. The measured bone strength indicators were Elastic modulus, Yield strength, and Maximum stress.

#### 2.5.3. Analysis of Bone Tissue Microstructure

The whole femur of the test chicken was scanned by a small animal CT imaging system (Micro-CT, ZKKS-MCTSharp, Zhongke Kaisheng Medical Technology Co., Ltd, Guangzhou, China) to analyze the microstructure of the femur. The measured microstructure metrics include: Bone mineral density (BMD), bone tissue ratio (BV/TV), bone surface area to bone volume ratio (BS/BV), bone surface area tissue volume ratio (BS/TV), trabecular bone thickness (Tb.Th), trabecular bone Number (Tb.N), and trabecular bone separation (Tb.Sp).

#### 2.5.4. Determination of Blood Biochemical Indicators

The blood biochemical indicators: Vitamin D receptor (VDR), Osteocalcin (OT), Osteoprotegerin (OPG), and Alkaline phosphatase (ALP) were determined according to the operation method provided by the ELISA kit (Baolai Biotechnology Co, Ltd, Guangzhou, China) manufacturer’s instruction manual.

#### 2.5.5. RNA Preparation and RT-qPCR

Seven bone metabolism-related genes (Appendix A) were selected for qRT-PCR. Total RNA was extracted from thoracic vertebrae and leg cartilage tissues. The total RNA reverse transcriptase kit (Takara, China) was used to synthesize cDNA. Real-time PCR ABI 7500 Fast Real-Time PCR System using SYBR premix Ex TaqTM II (Takara, Treasure Bioengineering Co., Ltd, Dalian, China) was used to perform real-time PCR. The 2^−ΔΔCT^ method was used to determine relative expression, and β-actin was used as the internal control for the normalization of the results.

### 2.6. Data Processing and Statistical Analysis

The primer design data analysis was performed using Oligo 6.0 and Primer Premer 5.0 software. The data processing software of the small animal CT imaging system was: ZKKS-Micro-CT 4.1. Excel 2013 (Microsoft Office, Washington, DC, USA) was used to process data, SPSS 21.0 (SPSS, Chicago, IL, USA) was used for statistical analysis, and the results were expressed as (mean ± standard deviation); Duncan’s method was used for significant difference analysis. The gene expression was calculated using the following formula:Ratio = 2^−ΔCTtarget(sample-calibrator)^/2^−ΔCTβ-actin (sample-calibrator)^

The sample represents the CT value of the sample in the experimental group, and the calibrator represents the CT value of the sample in the control group. The expression level of each sample target gene after normalization by β-actin treatment relative to the control group was calculated by the above formula.

## 3. Results

### 3.1. Comparative Analysis of Live Weight and Survival Rate

The addition of CHF and BPs to the diet had a certain effect on the growth of Cobb broilers (Table 4). At 30 and 60 days old, the live weight of the CHF group was higher than that of the NC group, while the live weight of the BPs group was lower than that of the NC group.At the age of 30 days, the CHF-H group was significantly higher than the BPs-L group, and the CHF-M group was significantly higher than the BPs-M and BPs-L groups (*p* < 0.05). At 60 days of age, the CHF group and NC group were significantly higher than the BPs group (*p* < 0.05). At 60 days of age, the survival rate of the CHF group was higher than that of the NC group, and the BPs group was lower than that of the NC group. In conclusion, the addition of CHF-H to the diet had the best effect on promoting the growth of Cobb broilers.

### 3.2. Comparative Analysis of Bone Strength Indexes

#### 3.2.1. Comparative Analysis of Modulus of Elasticity

The addition of CHF and BPs to the diet could increase the elastic modulus of the tibia in Cobb broilers (Figure 1B). By comparing the experimental group and the NC group at the same age, it can be seen that the elastic moduli of the tibia in the CHF group and the BPs group were higher than those in the NC group at the age of 30 days, but there was no significant difference (*p* > 0.05). At 60 days of age, the CHF-H and BPs groups were higher than the control group, but the difference was not significant (*p* > 0.05). Between different experimental groups at the same age: 30 and 60 days old, the CHF-H group had the largest elastic modulus, but there was no significant difference between the groups (*p* > 0.05). In the same experimental group at different ages: the elastic modulus of the tibia at the 30-day age was higher than that at the 60-day age, and the elastic modulus of the 30-day-old tibia in the CHF-L group was significantly higher than that at 60-day age (*p* < 0.05). At the age of 30 days, the elastic modulus of the CHF-H group was the highest, and that of the BPs-H group was the second highest. At 60 days of age, the elastic modulus of the tibia in the CHF-H group was the highest. In conclusion, CHF-H had the best effect on improving the elastic modulus of the tibia.

#### 3.2.2. Comparative Analysis of Yield Strength

The addition of CHF and BPs to the diets could increase the yield strength of the tibia in Cobb broilers (Figure 1C). Between the experimental group and the control group at the same age: for 30 days of age, the yield strength of the tibia in the CHF group, BPs-H, and BPs-M groups was higher than that in the NC group, but there was no significant difference (*p* > 0.05). For 60 days of age, the yield strength of the tibia in the CHF-H, CHF-M, and BPs groups was higher than that in the NC group, but the difference was not significant (*p* > 0.05). Between different treatment groups at the same age: for 30 days of age, the CHF-H group had the highest tibia yield strength. For 60 days of age, the yield strength of the BPs-H group was the highest, but there was no significant difference between the groups (*p* > 0.05). In the same treatment group at different ages: the yield strength of the tibia at 30 days was higher than that at 60 days, and the yield strength of the tibia at 30 days in CHF-H and CHF-L groups was significantly higher than that at 60 days (*p* < 0.05). At 30 days of age, tibia yield strength was highest in the CHF-H group and second in the CHF-L group. At 60 days of age, the tibial yield strength was the highest in the BPs-H group and second in the CHF-H group. In conclusion, CHF-H and BPs-H had the best effect on improving the yield strength of the tibia.

#### 3.2.3. Comparison and Analysis of Maximum Stress

The addition of CHF and BPs to the diet could affect the maximum stress of the tibia of Cobb broilers (Figure 1D). Between the experimental group and the control group at the same age: for 30 days of age, the maximum stress of the tibia in the CHF-H, CHF-M, and BPs-H groups was higher than that in the NC group, but there was no significant difference (*p* > 0.05). For 60 days of age, the maximum stress of the tibia in the CHF-H, CHF-M, and BPs groups was higher than that in the NC group, but the difference was not significant (*p* > 0.05). Between different treatment groups at the same age: for the 30-day-old, the maximum stress of the tibia was BPs-H group > CHF-H group > other experimental groups, and there was no significant difference between the experimental groups (*p* > 0.05). For the 60-day-old, the maximum stress of the tibia was BPs-M group > BPs-H group > CHF-H group > other experimental groups, and there was no significant difference between the experimental groups (*p* > 0.05). In the same treatment group at different ages: the maximum stress of the tibia at the 30-day age was higher than that at the 60-day age, and the yield strength of the tibia at the 30-day age in CHF-H and CHF-L groups was significantly higher than that at the 60-day age (*p* < 0.05). At 30 days of age, the maximum tibial stress in the BPS-H group was the highest, which was higher than that in the CHF and NC groups. At 60 days of age, the maximum tibial stress in the BPs-M group was the highest, which was higher than that in the CHF and NC groups. In conclusion, BPs had the best effect on increasing the maximum stress of the tibia.

### 3.3. Comparative Analysis of the Microstructure of Bone Tissue

The 3D structure of the distal cancellous bone of the Cobb broiler femur was observed from the small animal CT scans of the NC group and the experimental group (Figure 2). Compared with the NC group, it could be seen that CHF and BPs improved the bone tissue microstructure of the distal cancellous bone of the femur in Cobb broiler chickens to a certain extent, and from the longitudinal section and cross-section of the distal femoral cancellous bone, the CHF-H group compound was the best.

The microstructural parameters of the cancellous bone tissue of the distal femur of the 60-day-old experimental chicken were compared and analyzed (Figure 3). It was found that dietary supplementation of CHF and BPs improved the BMD and BV/TV of the distal cancellous bone of the femur of Cobb broilers (Figure 3A,B), and the BMD and BV/TV of the distal cancellous bone of the femur in the CHF and BPs groups were significantly higher than those in the NC group (*p* < 0.05). The addition of CHF and BPs to the diet decreased BS/BV and Tb.Sp in the distal cancellous bone of the femur of Cobb broilers (Figure 3C,F). The BS/BV and Tb.Sp of the distal cancellous bone of the femur in the CHF and BPs groups were lower than those in the NC group, but the difference was not significant (*p* > 0.05). The addition of CHF and BPs to the diet increased the BS/TV and Tb.N of the distal cancellous bone of the Cobb broiler femur (Figure 3D,G). The BS/TV and Tb.N of the distal cancellous bone of the femur in the CHF and BPs groups were higher than those in the NC group, but the difference was not significant (*p* > 0.05). The addition of CHF and BPs to the diet increased the trabecular bone thickness (Tb.Th) of the distal cancellous bone of the femur in Cobb broilers (Figure 3E), and the CHF-H group, CHF-M group, and BPs-H group were significantly higher than the NC group (*p* < 0.05). From the microstructure parameters of the cancellous bone of the distal femur, the BMD, BV/TV and BS/TV of Cobb broilers in the CHF group and BPs group were higher than those in the NC group, while BS/BV was lower than that in the NC group. These parameters indicated that the bone mass of the distal cancellous bone of the femur in the CHF group and the BPs group was higher than that in the NC group to a certain extent, and the CHF-H group had the highest bone mass. From the microstructural parameters of trabecular bone, the Tb.Th and Tb.N of the distal cancellous bone of the femur in the CHF group and the BPs group were higher than those in the NC group, while Tb.Sp was lower than that in the NC group. These parameters indicated that the number of bone trabeculae in the CHF and BPs groups was higher than that of the NC group, bone anabolism was also higher than that of the NC group to a certain extent, and the CHF-H group was the best.

### 3.4. Comparative Analysis of Blood Biochemical Indicators

#### 3.4.1. Comparative Analysis of VDR Content

The addition of CHF and BPs to the diet could increase the VDR content in the serum of Cobb broilers (Figure 4A). Between the experimental group and the control group at the same age: for 30 days of age, the serum VDR levels in the CHF group and the BPs group were higher than those in the NC group, and the BPs-H group was significantly higher than the NC group (*p* < 0.05). For the 60-day-old, the serum VDR levels in the CHF group and BPs group were higher than those in the NC group, and the CHF group was significantly higher than the NC group (*p* < 0.05). Between different experimental groups at the same age: for 30 days of age, the serum VDR content in the BPs-H group was significantly higher than that in the CHF-M and BPs-L groups (*p* < 0.05). For 60 days of age, the CHF-M group was significantly higher than BPs-M and BPs-L groups (*p* < 0.05). The same experimental group at different ages: the CHF group and BPs group showed that the serum VDR content at the 30-day age was higher than that at the 60-day age, and the serum VDR content of BPs-H and BPs-M groups at the 30-day age was significantly higher than that at the 60-day age (*p* < 0.05). In conclusion, the serum VDR content of the CHF group and BPs group was higher than that of the NC group at 30 and 60 days of age, and for 30 days of age, the serum VDR content of BPS-H was the highest. For the 60-day-old, serum VDR content in the CHF-M group was the highest. From the results, the CHF-M group had the highest VDR content in the serum.

#### 3.4.2. Comparative Analysis of OT Content

The addition of CHF and BPs to the diet affected the serum OT content of Cobb broilers (Figure 4B). Between the experimental group and the control group at the same age: for 30 days of age, the serum OT content of the CHF group and the BPs group was lower than that of the NC group, but the difference was not significant (*p* < 0.05). For the 60-day-old, the serum OT content of the CHF group and BPs group was lower than that of the NC group, and the serum OT content of the CHF-H and CHF-M group and BPs-H group was significantly lower than that of the NC group (*p* < 0.05). At the same age, among different experimental groups: the CHF-H group had the lowest serum OT content at 30 and 60 days of age, and there was no significant difference between the groups (*p* > 0.05). In the same experimental group at different ages: CHF-H, CHF-L, and BPs groups had lower serum OT levels at 30-day-old than 60-day-old. In addition, the 30-day-old serum OT content of the NC group and BPs-H group was significantly lower than that of the 60-day-old (*p* < 0.05). The serum OT content of the CHF group and BPs group was lower than that of the NC group at 30 and 60 days of age, and the CHF-H group had the lowest serum OT content, and it was the most significant in reducing the serum OT content.

#### 3.4.3. Comparative Analysis of OPG Content

The addition of CHF and BPs to the diet increased the OPG content in the serum of Cobb broilers (Figure 4C). Between the experimental group and the control group at the same age: for 30 days of age, the serum OPG levels in the CHF-H, CHF-M, and BPs-H, BPs-M groups were higher than those in the control group, and the BPs-H group was significantly higher than that in the NC group (*p* < 0.05). For 60 days of age, the serum OPG levels in CHF and BPs groups were higher than those in NC, and CHF-H, CHF-M, and BPs-H groups were significantly higher than those in the NC group (*p* < 0.05). Between different experimental groups at the same age: at the age of 30 days, the serum OPG content of the BPs-H group was the highest; at the age of 60 days, the serum OPG content of the CHF-H group was the highest. In the same treatment group at different ages: the OPG content in the serum of all treatment groups at 30 days of age was higher than that at 60 days of age, and the CHF-M group BPs-H, and BPs-M group 30-day-old serum OPG content was significantly higher than 60-day-old (*p* < 0.05). At 30 days of age, the serum OPG content in the experimental groups was higher than that in the NC group except for the BPs-L and CHF-L groups. At 60 days of age, the serum OPG content in the CHF and BPs groups was higher than that in the NC group. In conclusion, the BPs-H group had the best effect on increasing the OPG content in serum.

#### 3.4.4. Comparative Analysis of ALP Content

The addition of CHF and BPs to the diet could increase the ALP content in the serum of Cobb broilers (Figure 4D). Between the experimental group and the NC group at the same age: for 30 days of age, the serum ALP content of the CHF-H group, BPs-H, and BPs-M groups was higher than that of the NC group. For 60-day-old, the serum ALP levels in CHF and BPs groups were higher than those in the NC group, and CHF-H, BPs-H, and BPs-M groups were significantly higher than those in the NC group (*p* < 0.05). Different experiments at the same age: at 30 days of age, the serum ALP content of the BPs-H group was significantly higher than that of the CHF-M, CHF-L, and BPs-L groups (*p* < 0.05), and the CHF-H group was significantly higher than CHF-M, CHF-L and BPs-L groups (*p* < 0.05). At 60 days of age, the BPs-M group was significantly higher than the BPs-L group (*p* < 0.05). In the same experimental group at different ages: the serum ALP content of a 30-day-old was higher than that of a 60-day-old, and the serum ALP content of a 30-day-old in the CHF-H group was significantly higher than that of a 60-day-old (*p* < 0.05). The serum ALP content of the control group at 30 days of age was significantly higher than that at 60 days of age (*p* < 0.05). At 30 days of age, serum ALP content in BPs-H, CHF-H, and BPs-M groups was higher than that in the NC group. At 60 days of age, serum ALP content in CHF and BPs groups was higher than that in the NC group. In conclusion, the BPs-H group had the most obvious effect on increasing the ALP content in serum.

### 3.5. Comparative Analysis of Gene Expression Levels Related to Bone Metabolism

#### 3.5.1. Comparative Analysis of Relative Expression Levels of BMP-2

The addition of CHF and BPs to the diet could promote the expression of *BMP-2* mRNA in the thoracic tissue of Cobb broilers (Figure 5A). Between different experimental groups at the same age: at 30 days of age, the relative expression of *BMP-2* mRNA in the thoracic vertebrae of the CHF and BPs groups was higher than that of the NC group, and the BPs-H and BPs-M groups were significantly higher than those of the BPs-L group (*p* < 0.05). At 60 days of age, the relative expression of *BMP-2* mRNA in the CHF and BPs groups was higher than that in the NC group, and the BPs-H group was significantly higher than the BPs-L group (*p* < 0.05). In the same group at different ages: the relative expression of *BMP-2* mRNA in thoracic vertebra tissue at 30 days of age in the CHF group and BPs-H and BPs-M groups was higher than that at 60 days of age, the CHF-H, CHF-M, and BPs-M groups were significantly higher at 30 days than at 60 days (*p* < 0.05). 

The addition of CHF and BPs to the diet could promote the expression of *BMP-2* mRNA in the cartilage tissue of Cobb broiler chicken legs (Figure 5B). Between different experimental groups at the same age: for 30 days of age, the relative expression of *BMP-2* mRNA in leg cartilage tissue of the CHF group and BPs group was higher than that of the NC group, and the amount of BPs-H group was significantly higher than that of the CHF-L group and BPs-L group (*p* < 0.05). For 60 days of age, the relative expression of *BMP-2* mRNA in leg cartilage tissue of the CHF group and BPs group was higher than that of the NC group, and that of the BPs-H group was significantly higher than that of the BPs-L group (*p* < 0.05). In the same experimental group at different ages: the relative expression of *BMP-2* mRNA in the 30-day-old leg cartilage tissue of the CHF group and BPs group was higher than that of the 60-day-old, and the CHF-M group was significantly higher at 30 days than at 60 days (*p* < 0.05). 

In conclusion, the relative expression of *BMP-2* mRNA in the CHF-H group was only lower than that in the BPs-H and BPs-M groups in thoracic vertebrae and leg cartilage tissues at 30 and 60 days of age. The results showed that CHF also had the same ability to promote *BMP-2* mRNA expression as BPs. Among them, CHF-H had the strongest ability to promote *BMP-2* mRNA expression in the CHF group.

#### 3.5.2. Comparative Analysis of Relative Expression Levels of OPG

The addition of CHF and BPs to the diet could promote the expression of *OPG* mRNA in the thoracic tissue of Cobb broilers (Figure 5C). Between different experimental groups at the same age: for 30 days of age, the relative expression of *OPG* mRNA in the thoracic vertebrae of the CHF and BPs groups was higher than that of the NC group, and BPs-H group was significantly higher than the CHF-H group, CHF-L group and BPs-L group (*p* < 0.05), and the CHF-M group was significantly higher than the BPs-L group (*p* < 0.05). For 60 days of age, the relative expression of *OPG* mRNA in the thoracic vertebrae of the CHF and BPs groups was higher than that of the NC group, and the BPs-H and BPs-M groups were significantly higher than those of the CHF-L group (*p* < 0.05). In the same experimental group at different ages: the relative expression of *OPG* mRNA in the thoracic vertebra tissue of the CHF, BPs-H, and BPs-M groups at 30 days of age was higher than that of 60 days of age, and the CHF-M group at 30 days of age was significantly higher than that at 60 days of age. (*p* < 0.05).

The addition of CHF and BPs to the diet could promote the expression of *OPG* mRNA in the cartilage tissue of Cobb broiler chicken legs (Figure 5D). Different experiments at the same age: for 30 days of age, the relative expression of *OPG* mRNA in leg cartilage tissue of CHF and BPs groups was higher than that of the NC group, and that of the BPs-H group was significantly higher than that of the BPs-L group (*p* < 0.05). For 60 days of age, the relative expression of *OPG* mRNA in the leg cartilage tissue of the CHF and BPs groups was higher than that of the NC group, and the BPs-H group was significantly higher than that of the CHF-L group and BPs-L group (*p* < 0.05). In the same experimental group at different ages: the relative expression of *OPG* mRNA in the 30-day-old leg cartilage tissue of the CHF-M, CHF-L, and BPs-M groups was higher than that of the 60-day-old, but there was no significant difference between the treatment groups (*p* > 0.05). 

In conclusion, in the thoracic vertebrae and leg cartilage tissues, only the relative expression levels of *OPG* mRNA at 30-day and 60-day age in the BPs-H and BPs-M groups were higher than those in the CHF group, and CHF-H had the highest relative expression of OPG mRNA in the CHF group. The results indicated that CHF could promote the expression of *OPG* mRNA.

#### 3.5.3. Comparative Analysis of Relative Expression Levels of VDR

The addition of CHF and BPs to the diet could promote the *VDA* mRNA expression in the thoracic tissue of Cobb broilers (Figure 5E). Between different experimental groups at the same age: for 30 days of age, the relative expression of *VDR* mRNA in the thoracic vertebrae of the CHF and BPs groups was higher than that of the NC group, and the BPs-H group was significantly higher than the CHF-L group and BPs-M, BPs-L group (*p* < 0.05), CHF-H group was significantly higher than CHF-L group (*p* < 0.05). For 60 days of age, the relative expression of *VDR* mRNA in the CHF and BPs-H groups was higher than that in the NC group, and the CHF-H group was significantly higher than that in the BPs-L group (*p* < 0.05). In the same experimental group at different ages: the relative expression levels of *VDR* mRNA in the thoracic vertebra tissue of the CHF and BPs groups at the age of 30 days were higher than those at the age of 60 days, and the CHF-M group and the BPs-H group at the age of 30 days were significantly higher than those at the age of 60 days (*p* < 0.05).

The addition of CHF and BPs to the diet could affect the *VDR* mRNA expression in the cartilage tissue of Cobb broiler chicken legs (Figure 5F). Between different experimental groups at the same age: for 30 days of age, the relative expression of *VDR* mRNA in leg cartilage tissue of the CHF and BPs groups was lower than that of the NC group, and CHF-H and CHF-L groups were significantly higher than those of the BPs-M group (*p* < 0.05). For 60 days of age, the relative expression of *VDR* mRNA in leg cartilage tissue of the CHF group and BPs group was lower than that of the NC group, and the CHF-L group was significantly higher than the BPs-H and BPs-M group (*p* < 0.05), CHF-H group was significantly higher than the BPs-M group (*p* < 0.05). The same experimental group between different days: in the CHF-M group and BPs group at 30-day-old, the leg cartilage tissue *VDR* mRNA relative expression was higher than at 60-day-old, and the BPs-H group 30-day-old was significantly higher than at 60-day-old (*p* < 0.05). 

In conclusion, CHF and BPs could promote the expression of *VDR* mRNA in the thoracic vertebra of Cobb broilers and inhibit the expression of *VDR* mRNA in the leg cartilage tissue.

#### 3.5.4. Comparative Analysis of the Relative Expression of Runx-2

The addition of CHF and BPs to the diet could promote the expression of *Runx-2* mRNA in the thoracic tissue of Cobb broilers (Figure 5G). Between different experimental groups at the same age: for 30 days of age, the relative expression of *Runx-2* mRNA in the thoracic vertebra tissue of the CHF and BPs groups was higher than that of the control NC group, and the relative expression of *Runx-2* mRNA in thoracic vertebra tissue of the BPs-H group was significantly higher than CHF-L, BPs-L group (*p* < 0.05), CHF-M group was significantly higher than CHF-L group (*p* < 0.05). For 60 days of age, the relative expression of *Runx-2* mRNA in the CHF and BPs groups was higher than that in the NC group, but there was no significant difference between the experimental groups (*p* > 0.05). The same experimental group between different days and ages: the relative expression levels of *Runx-2* mRNA in thoracic vertebra tissue at 30 days of age in CHF-H, CHF-M, and BPs groups were higher than those at 60 days of age, and the CHF-H, CHF-M group, and BPs-H group were significantly higher than 60-day-old at 30-day age (*p* < 0.05).

The addition of CHF and BPs to the diet promoted the expression of *Runx-2* mRNA in the cartilage tissue of Cobb broiler chicken legs (Figure 5H). Between different experimental groups at the same age: for 30 days of age, the relative expression of *Runx-2* mRNA in leg cartilage tissue of the CHF and BPs groups was higher than that of the NC group, and the BPs-H group was significantly higher than that of the BPs-M group (*p* < 0.05). For 60 days of age, the relative expression of *Runx-2* mRNA in leg cartilage tissue of the CHF group and BPs group was higher than that of the control group, and the BPs-H group was significantly higher than that of the CHF-L group and BPs-L group (*p* < 0.05). In the same experimental group at different ages: the relative expression of *Runx-2* mRNA in the 30-day-old leg cartilage tissue in the CHF, BPs-H, and BPs-L groups was higher than that in the 60-day-old group, but there was no significant difference among the groups (*p* > 0.05). 

In conclusion, in thoracic vertebrae and leg cartilage tissues, the relative expression levels of *Runx-2* mRNA at 30 and 60 days of age in the BPs-H group and CHF-H group were higher than those in other groups. The results showed that the CHF-H group and the BPs-H group could promote the expression of *Runx-2* mRNA.

#### 3.5.5. Comparative Analysis of Relative Expression Levels of RANK

The addition of CHF and BPs to the diet could inhibit the expression of *RANK* mRNA in the thoracic tissue of Cobb broilers (Figure 5I). Between different experimental groups at the same age: for 30 days of age, the relative expression of *RANK* mRNA in the thoracic vertebrae of the CHF and BPs groups was lower than that of the NC group, and the BPs-M group was significantly higher than that of the CHF-H and BPs-H groups (*p* < 0.05). For 60 days of age, the relative expression of *RANK* mRNA in the CHF and BPs groups was lower than that in the NC group, and the CHF-H group was significantly lower than that in the BPs-L group (*p* < 0.05). In the same experimental group at different ages: the relative expression of *RANK* mRNA in thoracic vertebra tissue at 30 days of age in both the CHF and BPs groups was lower than that at 60 days of age, and the CHF-H group and the BPs-H and BPs-L groups were significantly lower at 30 days of age than at 60 days of age (*p* < 0.05). The results showed that the relative expression of *RANK* mRNA in thoracic vertebra tissue increased with the increase in age. 

The addition of CHF and BPs to the diet could inhibit the expression of *RANK* mRNA in the cartilage tissue of Cobb broiler chicken legs (Figure 5J). Between different experimental groups at the same age: for 30 days of age, the relative expression of *RANK* mRNA in the leg cartilage tissue of the CHF and BPs groups was lower than that of the NC group. The CHF-L group was significantly higher than the CHF-H, CHF-M, and BPs-H groups (*p* < 0.05). For 60 days of age, the relative expression of *RANK* mRNA in leg soft tissue of the CHF and BPs groups was lower than that of the NC group, and the CHF-M group was significantly lower than that of the CHF-L group (*p* < 0.05). In the same experimental group at different ages: in the CHF group and BPs-H, BPs-L groups all 30-day-old leg cartilage tissue relative expression of *RANK* mRNA was lower than 60-day-old, and the 30-day age of the BPs-H group was significantly lower than that of the 60-day age (*p* < 0.05). The results showed that the relative expression of *RANK* in leg cartilage tissue increased with the increase in age. 

In conclusion, in thoracic vertebrae and leg cartilage tissues, the relative expression of *RANK* mRNA in the BPs-H group and CHF-H group was lower than that in other groups at 30 and 60 days of age, and the CHF-H group was lower than in the BPs-H group. The results indicated that CHF-H had a better effect than BPs-H in inhibiting the expression of *RANK* mRNA.

#### 3.5.6. Comparative Analysis of Relative Expression Levels of RANKL

The addition of CHF and BPs to the diet could inhibit the expression of *RANKL* mRNA in the thoracic tissue of Cobb broilers (Figure 5K). Between different experimental groups at the same age: for 30 days of age, the relative expression of *RANKL* mRNA in the thoracic vertebrae of the CHF and BPs groups was lower than that of the NC group, and the BPs-L group was significantly higher than that of the CHF-H and BPs-H groups (*p* < 0.05), the CHF-H group was significantly lower than the CHF-L group (*p* < 0.05), and the BPs-H group was significantly lower than the CHF-L group (*p* < 0.05). For 60 days of age, the relative expression of *RANKL* mRNA in the CHF and BPs groups was lower than that in the NC group, and the CHF-L group was significantly higher than the BPs-H and CHF-H groups (*p* < 0.05), the BPs-H group was significantly lower than the BPs-L group and CHF-L group (*p* < 0.05). In the same experimental group at different ages: the relative expression of *RANKL* mRNA in thoracic vertebra tissue at 30 days of age in both CHF and BPs groups was lower than that at 60 days of age, the BPs-L group was significantly lower at the age of 30 days than at the age of 60 days (*p* < 0.05). The results showed that the relative expression of *RANKL* mRNA in thoracic vertebra tissue increased with the increase in age.

The addition of CHF and BPs to the diet could inhibit the expression of *RANKL* mRNA in the cartilage tissue of Cobb broiler chicken legs (Figure 5L). Between different experimental groups at the same age: for 30 days of age, the relative expression of *RANKL* mRNA in leg cartilage tissue of CHF and BPs groups was lower than that of the NC group, and the CHF-L group was significantly higher than that of the BPs group (*p* < 0.05). For 60 days of age, the relative expression of *RANKL* mRNA in leg cartilage tissue of CHF and BPs groups was lower than that of the control group, and the CHF-L group was significantly higher than the BPs-M and BPs-L groups (*p* < 0.05). The same experimental group between different days and ages: the relative expression of *RANKL* mRNA in 30-day-old leg cartilage tissue in the CHF-H, CHF-L group, and the BPs-H, BPs-L group was lower than that in the 60-day-old group. The results showed that the relative expression of *RANKL* mRNA in leg cartilage tissue increased with the increase in age. 

In conclusion, in thoracic vertebrae and leg cartilage tissues, the CHF-H group had the best effect of inhibiting *RANKL* mRNA expression in the CHF group at the age of 30 and 60 days.

#### 3.5.7. Comparative Analysis of Relative Expression Levels of TNF-α

The addition of CHF and BPs to the diet could inhibit the expression of *TNF-α* mRNA in the thoracic tissue of Cobb broilers (Figure 5M). Between different experimental groups at the same age: for 30 days of age, the relative expression of *TNF-α* mRNA in the thoracic vertebrae of the CHF and BPs groups was lower than that of the NC group, and the CHF-M group was significantly lower than that of the CHF-L group (*p* < 0.05). For 60 days of age, the relative expression of *TNF-α* mRNA in the CHF and BPs groups was lower than that in the NC group, and the CHF-H group was significantly lower than that in the BPs-L group and the CHF-L group (*p* < 0.05). The same experimental group between different days and ages: the relative expression of *TNF-α* mRNA in thoracic tissue at 30 days of age in the CHF group and the BPs-H and BPs-L groups was lower than that at 60 days of age, and in the BPs-L group at 30 days of age was significantly lower than that at 60 days of age (*p* < 0.05).

The addition of CHF and BPs to the diet could inhibit the expression of *TNF-α* mRNA in the cartilage tissue of Cobb broiler chicken legs (Figure 5N). Between different experimental groups at the same age: for 30 days of age, the relative expression of *TNF-α* mRNA in the leg cartilage tissue of the CHF and BPs groups was lower than that of the NC group. The CHF-L group was significantly higher than the CHF-H, CHF-M, and BPs-M groups (*p* < 0.05), and the BPs-M group was significantly lower than the BPs-L group (*p* < 0.05). For 60 days of age, the relative expression of *TNF-α* mRNA in the leg cartilage tissue of the CHF and BPs groups was lower than that of the NC group, and that of the CHF-L group was significantly higher than that of the BPs-H and BPs-M groups (*p* < 0.05). The same experimental group between different days and ages: the relative expression of *TNF-α* mRNA in the 30-day-old leg cartilage tissue of the CHF group and BPs-H group was lower than that of the 60-day-old group, and the CHF-H and CHF-M groups were significantly lower than the 60-day-old group (*p* < 0.05). 

In conclusion, in thoracic vertebrae and leg cartilage tissues, the CHF-H group had the best effect of inhibiting the expression of *TNF-α* mRNA in the CHF group at the age of 30 and 60 days.

## 4. Discussion

Body weight and survival rate are important phenotypic indicators reflecting the individual health status of animals. This experimental study showed that the body weight at 30 and 60 days in the CHF group was higher than that of the NC group, and while the live weight of the BPs group was lower than that of the NC group, the results showed that CHF had good safety and was better than BPs in promoting growth.

In the process of growth and development, the order of tissue development is bone, muscle, fat, etc. [13]. As the fulcrum of the development of the whole body, bones play an important role in the protection, support, and hematopoiesis of the body. The development of poultry skeletons has an important impact on their growth and production performance [14].

Bone biomechanics is the study of the mechanical properties of bone tissue under the action of external force and the biological effect of bone after the action of force, which can directly reflect the strength and toughness of bone [15]. This study showed that the elastic modulus, yield strength, and maximum stress of the tibia in the CHF-H group and BPs-H group were higher than those in the NC group at 30 and 60 days of age. The results indicated that CHF could improve the tibia bone strength of Cobb broilers, and its effect was not weaker than that of BPs. This study also showed that the three indexes of elastic modulus, yield strength, and maximum stress of the tibia were higher at 30-day-old than 60-day-old. The results of this study are consistent with those of Jensen et al. and Curtis et al. [16,17]. In childhood, bone collagen content is high and its elasticity is large, so it is not easy to fracture but it is easy to deform. In adulthood, as the deposition of calcium salts increases, the glial part of the bone gradually decreases, and the brittleness of the bone increases. In old age, bone loss increases and bone strength decreases.

Micro CT has been considered the gold standard for evaluating bone morphology and bone microarchitecture [18,19]. Micro CT can accurately quantify bone parameters such as BMD, Tb.Th, Tb.N, Tb.Sp and other bone parameters. In recent years, many scholars have applied Micro CT technology to the study of chicken bones. In this experiment, the femur bone tissue of Cobb broilers was detected by Micro-CT scanning. The results showed that the addition of CHF and BPs to the feed improved the bone tissue microstructure of the distal cancellous bone of the femur in broilers to a certain extent compared with the NC group. The BMD, BV/TV, BS/BV, Tb.Th and Tb.N of the distal cancellous bone of the femur in the CHF and BPs groups were higher than those in the NC group, while BS/TV and Tb.Sp were lower than those in the NC group. BMD represents the bone mineral density per unit area of bone tissue, which is a very important indicator to reflect bone metabolism. BV/TV refers to the ratio of bone tissue volume to tissue volume, which can directly reflect changes in bone mass and is a common indicator for evaluating bone mass. BS/BV and BS/TV can indirectly reflect the amount of bone mass. For the microstructural analysis of trabecular bone, Tb.Th, Tb.N, and Tb.Sp are the main indicators used to evaluate the spatial morphological structure of trabecular bone. When osteoporosis occurs, bone catabolism is greater than bone anabolic metabolism. At this time, the values of Tb.Th and Tb.N will decrease, and the value of Tb.Sp will increase. The results showed that the CHF and BPs used in this experiment could improve the microstructure of the bone tissue, increase bone density, and improve the microstructure of trabecular bone in the test chickens. Chai et al. [20] found that Gushukang capsules made from Chinese herbal medicines can promote bone formation, inhibit bone resorption, and increase bone density in patients with osteoporosis. Fernández-Martín et al. [21] found that alendronate treatment could increase the subchondral bone volume fraction and improve the microstructure of subchondral cancellous bone in guinea pigs with osteoarthritis. Wu et al. [22] found that Xianling Gubao Capsules can prevent the occurrence of osteoporosis in ovariectomized rats, improve bone microstructure, make trabecular bone appear thick, dense and uniform, and maintain bone structure and strength during the period of drug withdrawal. It can be seen that sodium bisphosphonate and CHF can improve bone microstructure.

The dynamic balance of some important indicators in the blood in the process of bone metabolism is the key factor to ensuring the normal growth and development of bones. VDR is an important molecule that regulates bone development, metabolism, and homeostasis, and plays an important role in maintaining calcium and phosphorus metabolism in the body and regulating cell proliferation and differentiation. It can promote the synthesis of osteopontin and osteocalcin, and participate in the formation and mineralization of bone [23]. In this experiment, the serum VDR levels in the CHF group and BPs group at 30 and 60 days of age were higher than those in the NC group, and the serum VDR levels at 30 days of age were higher than those at 60 days of age. It showed that both CHF and BPs have the effect of promoting bone formation. This was consistent with the research of Udagawa et al. [24]. When the VDR content increased, it inhibited bone resorption, promoted osteocalcin synthesis, and participated in bone formation and mineralization. 

OT is a marker of mature osteoblasts, and its production increases with mineralization and osteoblast differentiation [25]. In this experiment, the serum OT content of the 30- and 60-day-old CHF group and BPs group was lower than that of the NC group, the CHF-H group had the lowest serum OT content, and the 30-day-old serum OT content was lower than the 60-day-old. By comparing the content of OT, it was found that the NC group was higher than the CHF group and the BPs group. It could be concluded that CHF and BPs have the effect of inhibiting the release of OT into the blood, thereby inhibiting bone transformation. 

The decrease of OPG and the increase of PTH in serum could be used as the markers of the decrease in bone mineralization [26,27]. Proper OPG supplementation can prevent the occurrence of osteoporosis in ovariectomized rats [28]. OPG is also involved in the process by which various hormones and cytokines regulate osteoclastogenesis. In the bone tissue microenvironment, it is an important regulatory molecule for osteoblasts and bone marrow stromal cells to regulate the differentiation and activation of osteoclasts [29]. In this experiment, the content of OPG in the 30-day-old serum of the CHF-H and CHF-M group, and BPs-H and BPs-M group was higher than that of the NC group. The serum OPG content in the CHF group and BPs group was higher than that in the NC group at 60 days of age. The OPG content in the serum of a 30-day-old was higher than that of a 60-day-old. It showed that both CHF and bisphosphonates have the effect of promoting bone formation and inhibiting bone resorption. This was consistent with the findings of Bennett et al. [28], which showed that OPG in serum decreased, bone formation decreased, and bone resorption increased, when OPG was elevated, bone formation increased, bone resorption decreased, and bone increased, and the results indicated that CHF and BPs have bone-enhancing effects. 

ALP is a marker of osteoblast maturation [30]. In this experiment, at 30 days of age, the serum ALP content of the CHF-H group and BPs-H and BPs-L groups was higher than that of the NC group, and at 60 days of age, the serum ALP content of the CHF group and BPs group was higher than that of the control group. It shows that both CHF and BPs can promote osteogenesis. The serum ALP content of a 30-day-old was higher than that of a 60-day-old, which was also consistent with the bone strength indexes (elastic modulus, yield strength, maximum stress) of a 30-day-old being higher than a 60-day-old.

In recent years, in the study of functional genes related to bone characteristics of chickens, many genes were found to be involved in the regulation of bone growth and development and the occurrence of bone diseases. *BMP-2* is an initiating factor for bone growth. *BMP-2* can not only promote the differentiation of osteoblasts but also direct the differentiation of osteoblast precursor cells into osteoblasts [31,32]. Chugh et al. [33] found that *BMP-2* could induce mouse bone marrow-derived cells W-20-17 to exhibit an osteoblast phenotype. Liu et al. [34] injected a human-transfected *BMP-2* plasmid into the lumbar intervertebral space of adult New Zealand rabbits. After six weeks, CT scans and histological examinations found that new bone was formed at the injection site. Primadhi et al. [35] reported that *BMP-2* has a strong proliferation and expression effect on most bone matrix proteins, and can mineralize human osteoblasts. In this experiment, in the thoracic vertebrae and leg cartilage tissues, the relative expression levels of *BMP-2* mRNA in the thoracic vertebrae of the CHF and BPs groups were higher than those of the NC group at 30 and 60 days of age, and the relative expression of *BMP-2* mRNA in the CHF-H group was only lower than that in BPs-H and BPs-M groups. It showed that CHF also had the ability to promote the expression of *BMP-2* mRNA like BPs, and CHF-H in the CHF group had the strongest ability to promote the expression of *BMP-2* mRNA. It indicated that CHF could promote the expression of *BMP-2*, induced bone mesenchymal stem cells to differentiate into osteoblasts, and promoted bone formation. 

*OPG* is known as an osteoclast inhibitory factor and has the function of reducing osteoclast differentiation and increasing bone density [36]. *OPG* can reflect the level of bone metabolism. When *OPG* is elevated, bone formation is increased, and when *OPG* is decreased, bone formation is decreased [37]. *OPG* gene knockout resulted in a significant reduction in bone mass in mice. Overexpression of *OPG* resulted in a decrease in the number of osteoclasts and an increase in bone mass [38]. In this experiment, in the thoracic vertebrae and leg cartilage tissues, the relative expression of *OPG* mRNA in the thoracic vertebrae of the CHF and BPs groups was higher than that of the NC group at the age of 30 and 60 days. In the experimental group, only the relative expression of *OPG* mRNA in the BPs-H and BPs-M groups was higher than that in the CHF group, and the relative expression of *OPG* mRNA in CHF-H was the highest in the CHF group. It showed that CHF can promote the expression of *OPG* mRNA, inhibit the differentiation of osteoclasts and reduce bone resorption. 

The action of *VDR* on bone tissue is bidirectional. *VDR* on osteoblasts can regulate *VDR* on osteoclasts, inhibit their proliferation, and promote their differentiation. Therefore, it has a bidirectional regulation of bone synthesis and catabolism [39,40]. In this experiment, in the thoracic spine tissue, the relative expression of *VDR* mRNA in the 30-day-old CHF group and BPs group was higher than that in the NC group, and the relative expression of *VDR* mRNA in the 60-day-old CHF group and BPs-H group was higher than that in the NC group. In leg soft tissue, the relative expression of *VDR* mRNA in leg cartilage tissue of the 30- and 60-day-old CHF group and BPs group was lower than that of the NC group. The results indicated that CHF and BPs could promote the expression of *VDR* mRNA in the thoracic vertebrae of Cobb broilers and inhibit the expression of *VDR* mRNA in the leg cartilage tissue. 

As a specific transcription factor of osteocytes, *Runx-2* initiates and regulates the differentiation of osteoblasts, and plays an important role in the formation and reconstruction of bone tissue. It regulates the differentiation of mesenchymal stem cells into osteoblasts and promotes the maturation of chondrocytes. It is an essential gene for bone development and bone formation and plays a decisive role in the development of osteoblasts and osteoclasts [41,42]. BMP-2/Smads/Runx-2 signaling pathway is an important pathway involved in osteoblast differentiation and osteoblast extracellular matrix synthesis and promotes bone formation. BMP-2 is an important extracellular signaling molecule that promotes bone formation and induces osteoblast differentiation. BMP-2 binds to cell surface type II receptors to form dimers. Type I receptors specifically bind to this dimer and undergo autophosphorylation, which in turn phosphorylates Smad1 and Smad5 to activate them. Then it binds to Smad4 to form a complex, enters the nucleus, and interacts with *Runx-2* to participate in osteoblast phenotype gene expression and differentiation [43]. In this experiment, in thoracic vertebrae and leg cartilage tissues, the relative expression of *Runx-2* mRNA in thoracic vertebrae in the 30- and 60-day-old CHF group and BPs group was higher than that in the NC group. In conclusion, in thoracic vertebrae and leg cartilage tissues, the relative expression levels of *Runx-2* mRNA in the BPs-H group and CHF-H group were higher than those in other groups at 30 and 60 days of age. It showed that the CHF-H group and the BPs-H group can promote the expression of *Runx-2* mRNA, promote the differentiation of bone mesenchymal stem cells into osteoblasts, and promote bone formation. 

*RANK* is also a member of the TNF receptor family. It is the signal receptor of *RANKL* and is a type I transmembrane protein. *RANKL* is known as a tumor necrosis factor activation-inducing cytokine (TRANCE), osteoclast differentiation factor (ODF), and osteoprotegerin ligand (OPGL). *RANKL* can not only bind to the functional receptor-*RANK* but also bind to the decoy receptor-*OPG.* The OPG/RANK/RANKL system plays a key regulatory role in maintaining the dynamic balance between bone resorption and bone formation, preventing bone loss, and ensuring normal bone turnover and renewal [44,45]. Its mechanism of action is that osteoclasts (OC) and osteoblasts (OB) mediate the process of bone resorption and synthesis, respectively. *RANKL* is mainly expressed by osteoblast precursor cells and bone marrow stromal cells, while *RANK* is mainly expressed by osteoclasts. The combination of *RANKL* and *RANK* can promote osteoclast differentiation, fusion, and maturation, and inhibit osteoclast apoptosis, thereby promoting bone resorption. At the same time, osteoblasts also express OPG, which is essentially a receptor decoy that can competitively bind to the *RANKL* family to inhibit the formation and maturation of osteoclasts. When *RANK* competes with OPG for binding to *RANKL*, bone resorption and bone formation are in equilibrium [46,47]. In this experiment: the relative expression of *RANK* and its receptor *RANKL* mRNA in the thoracic spine and leg cartilage tissue at the age of 30 and 60 days in the CHF group and the BPs group was lower than that in the NC group. This indicated that CHF and BPs had inhibitory effects on the mRNA expression of *RANK* and *RANKL*, and concluded that CHF could inhibit the activity of osteoclasts and inhibit the process of bone resorption. The effect of CHF-H was the most obvious in the CHF group. 

*TNF-α* is produced by activated macrophage cells and can be used as an apoptosis factor, an inflammation-promoting factor, an important osteoclast activating factor, and an important regulator of bone metabolism and remodeling. It can stimulate the differentiation of osteoclasts in a synergistic manner and is a potent bone resorption factor [48,49]. *TNF-α* can reduce the special markers of osteoblast differentiation, and can also regulate the expression of *BMP-2*, and finally inhibit the differentiation of osteoblasts. It also promotes the mitosis of osteoclast precursor cells and the differentiation and maturation of osteoclast progenitor cells, and can also inhibit the apoptosis of osteoclasts and inhibit bone remodeling, ultimately leading to bone loss [50,51]. In this experiment: in the thoracic vertebrae and leg cartilage tissues, the relative expression levels of *TNF-α* mRNA in the 30- and 60-day-old CHF group and BPs group were lower than those in the NC group. In this experiment, the lower the *TNF-α* expression in the experimental group, the higher the bone strength index (elastic modulus, yield strength, maximum stress). This indicated that both CHF and BPs could inhibit *TNF-α* expression and promote osteogenic differentiation. The higher the *TNF-α* expression level, the lower the osteoblast activity, the weaker the bone formation, and the worse the bone quality. The CHF-H group had the best effect of inhibiting the expression of *TNF-α* mRNA in the CHF group.

In this study, the experimental chickens were provided with different doses of CHF and BPs in their feed. The 30- and 60-day-old live weight, tibial bone strength, the microstructure of the distal femoral cancellous bone, blood biochemical indexes related to bone metabolism, and genes related to bone metabolism were detected and analyzed. The results showed that adding CHF and BPs to the diet could improve the bone properties of broilers; the bone strength of broilers decreased with the increase in age. CHF and BPs improved the bone tissue microstructure of the distal femoral cancellous bone to a certain extent, increased the BMD, BV/TV, BS/TV, Tb.Th and Tb.N of the distal femoral cancellous bone, and decreased the BS/BV, Tb.Sp. CHF and BPs could reduce the content of OT by increasing the content of VDR, OPG, and ALP in serum. In addition, CHF and BPs upregulated the expression of osteogenic genes (BMP-2, OPG, Runx-2), which promoted bone formation and also downregulated the expression of osteolytic genes (RANK, RANKL, TNF-α), which inhibited bone resorption, thus affecting bone metabolism and skeletal properties.

## 5. Conclusions

CHF could improve the bone properties of Cobb broilers, and its effect is not weaker than that of BPs. From the indicators of live weight and bone properties, the CHF-H group was the best formula for improving the bone properties of Cobb broilers. It could maintain the dynamic balance between bone resorption and bone formation by regulating the OPG/RANK/RANKL system, and promoting the expression of *BMP-2* and *Runx-2* in BMP-2/Smads/Runx-2, an important pathway for bone formation. At the same time, it inhibited the expression of the *TNF-α* gene to improve the bone characteristics of Cobb broilers.

## Figures and Tables

**Figure 1 genes-13-01865-f001:**
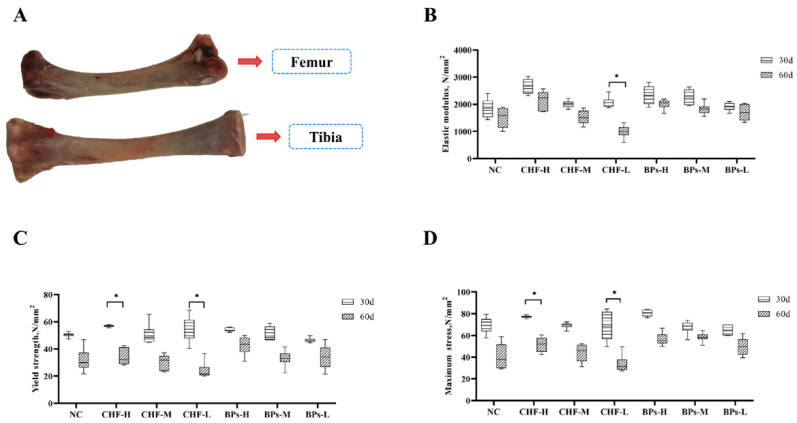
Comparison of indicators related to tibial bone strength. (**A**) Femur and tibia at 60 days of age. (**B**) Comparison of elastic moduli of the tibia. (**C**) Comparison of tibia yield strength. (**D**) Comparison of maximum tibia stress. * indicates significant difference (*p* < 0.05).

**Figure 2 genes-13-01865-f002:**
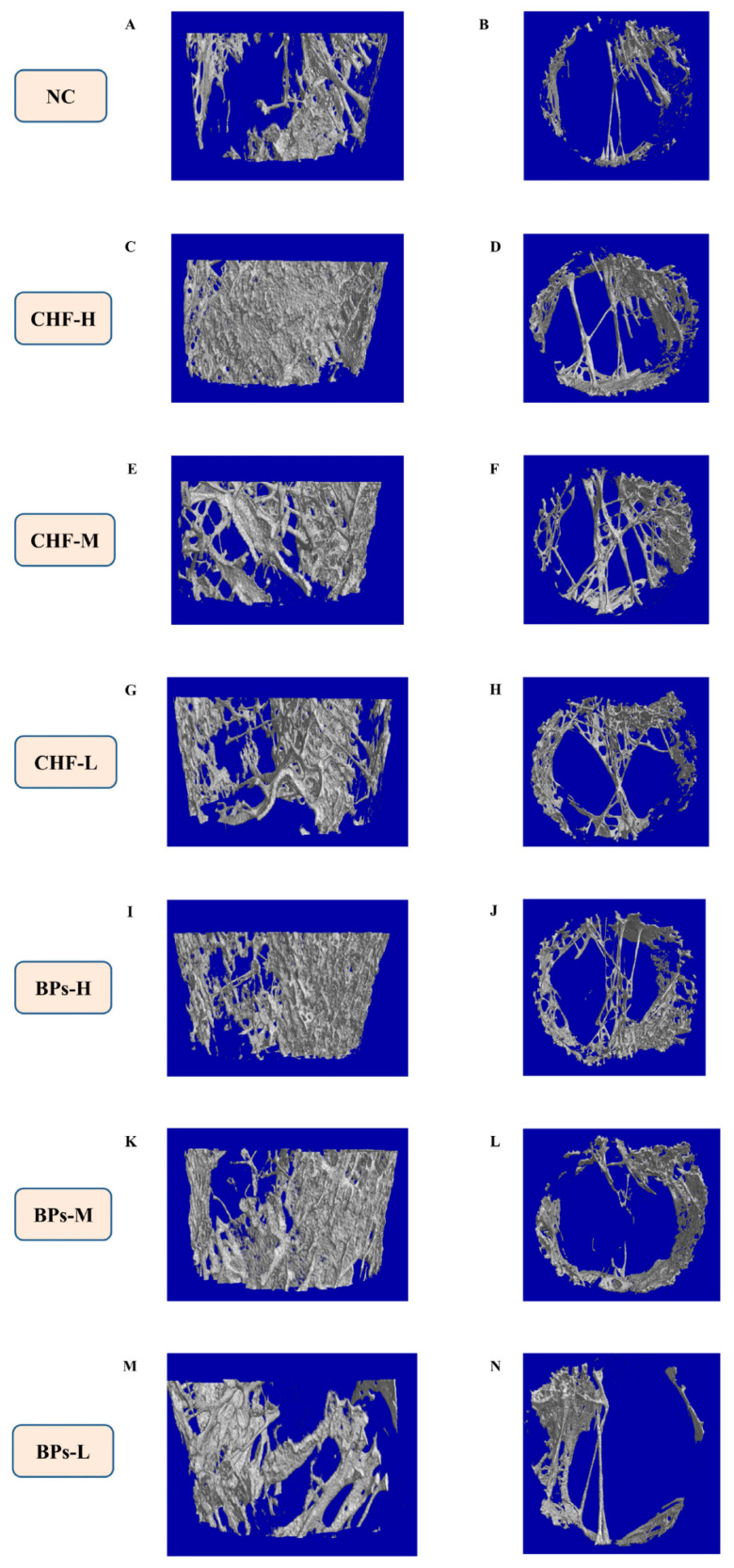
3D structure of cancellous bone of distal femur at 60 days of age. (**A**) Longitudinal section of the distal cancellous bone of the femur in the NC group. (**B**) Cross-section of the distal cancellous bone of the femur in the NC group. (**C**) Longitudinal section of distal femur cancellous bone in CHF-H group. (**D**) Cross-section of distal femur cancellous bone in CHF-H group. (**E**) Longitudinal section of distal femur cancellous bone in CHF-M group. (**F**) Cross-section of distal femur cancellous bone in CHF-M group. (**G**) Longitudinal section of distal femur cancellous bone in CHF-L group. (**H**) Cross-section of distal femur cancellous bone in CHF-L group. (**I**) Longitudinal section of distal femur cancellous bone in BPs-H group. (**J**) Cross-section of distal femur cancellous bone in BPs-H group. (**K**) Longitudinal section of the distal cancellous bone of the femur in the BPs-M group. (**L**) Cross-section of the distal femur cancellous bone in the BPs-M group. (**M**) Longitudinal section of distal femur cancellous bone in BPs-L group. (**N**) Cross section of distal femur cancellous bone in BPs-L group.

**Figure 3 genes-13-01865-f003:**
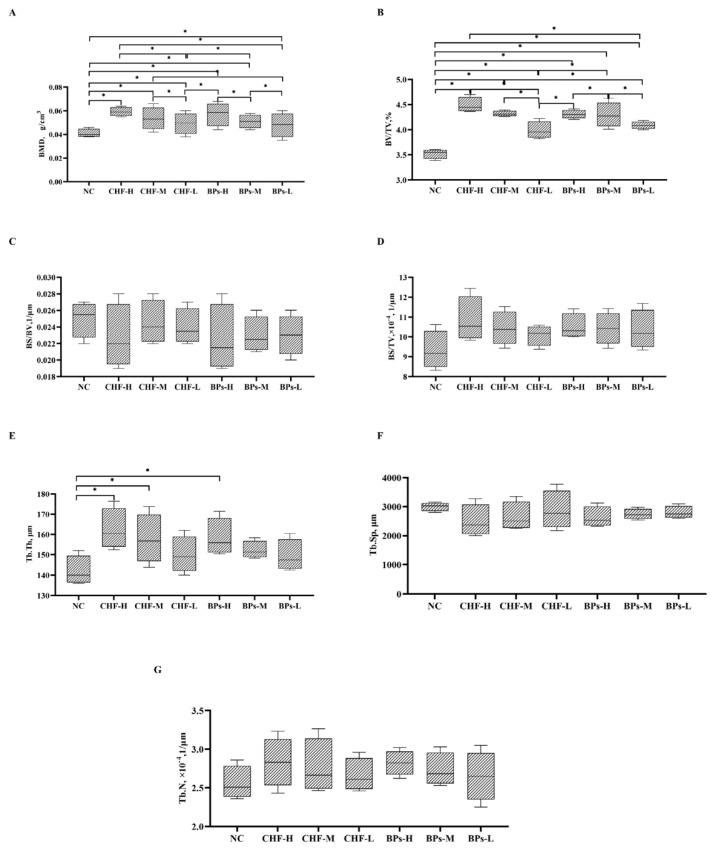
Comparison of microstructural parameters of cancellous bone tissue of distal femur at 60 days of age. (**A**) BMD of cancellous bone of distal femur. (**B**) BV/TV of cancellous bone of distal femur. (**C**) BS/BV of cancellous bone of distal femur. (**D**) BS/TV of cancellous bone of distal femur. (**E**) Tb.Th of cancellous bone of distal femur. (**F**) Tb.N of cancellous bone of distal femur. (**G**) Tb.Sp of cancellous bone of distal femur. * indicates significant difference (*p* < 0.05).

**Figure 4 genes-13-01865-f004:**
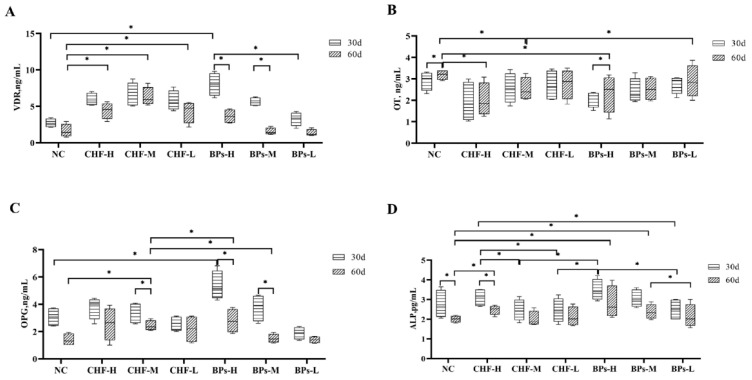
Comparison of biochemical indicators related to bone metabolism in serum. (**A**) Comparison of the VDR content in serum. (**B**) Comparison of OT content in serum. (**C**) Comparison of OPG levels in serum. (**D**) Comparison of ALP levels in serum. * indicates significant difference (*p* < 0.05).

**Figure 5 genes-13-01865-f005:**
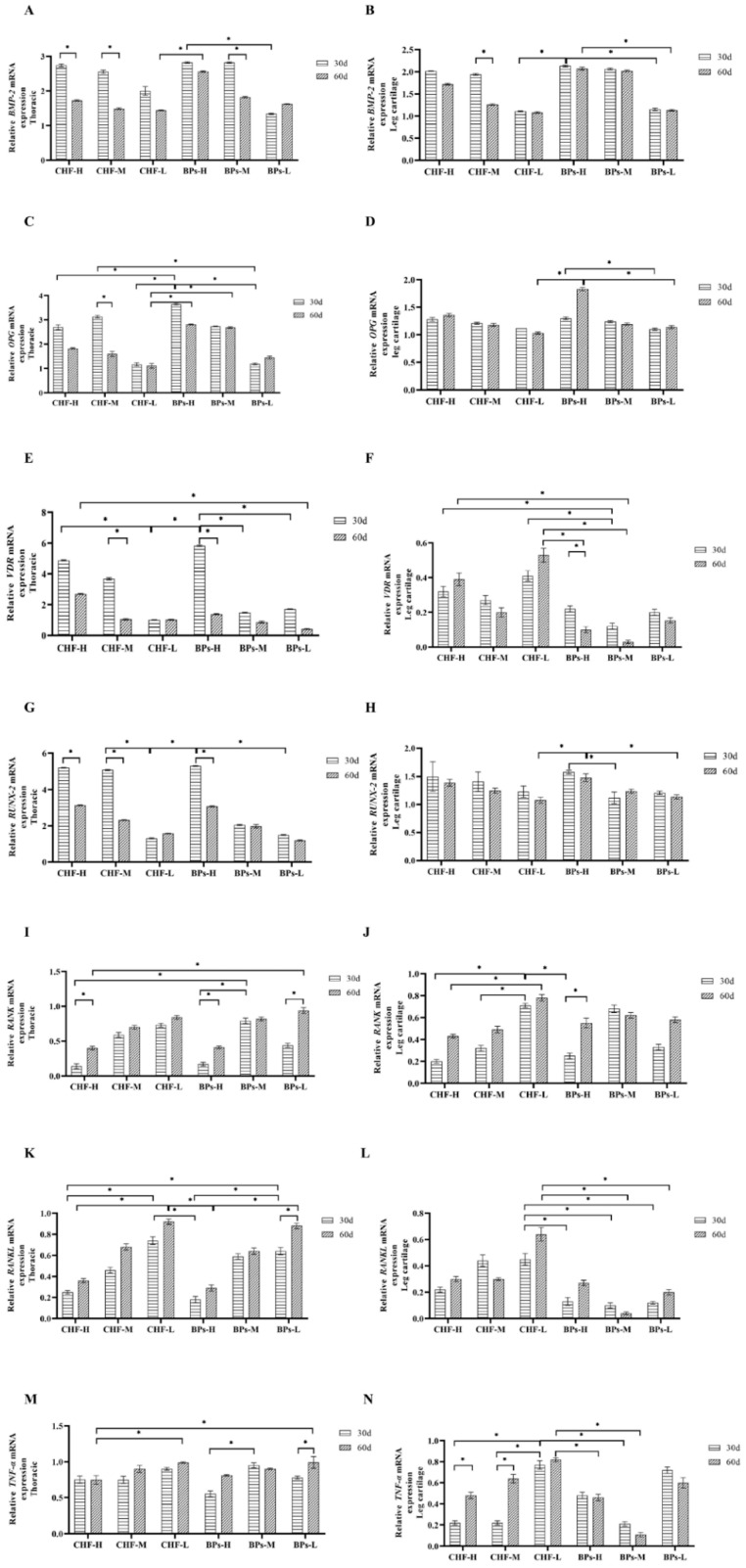
Comparison of relative mRNA expression levels of bone metabolism-related genes in thoracic vertebrae and leg cartilage tissues. (**A**) Comparison of the relative mRNA expression levels of *BMP-2* in thoracic vertebrae. (**B**) Comparison of the relative mRNA expression levels of *BMP-2* in leg cartilage tissues. (**C**) Comparison of the relative mRNA expression levels of *OPG* in thoracic vertebrae. (**D**) Comparison of the relative expression levels of *OPG* mRNA in leg cartilage tissue. (**E**) Comparison of the relative mRNA expression levels of *VDR* in thoracic vertebrae. (**F**) Comparison of the relative mRNA expression levels of *VDR* in leg cartilage tissue. (**G**) Comparison of the relative mRNA expression levels of *RUNX-2* in thoracic vertebrae. (**H**) Comparison of the relative mRNA expression levels of *RUNX-2* in leg cartilage tissue. (**I**) Comparison of the relative mRNA expression levels of *RANK* in thoracic vertebrae. (**J**) Comparison of the relative mRNA expression levels of *RANK* in leg cartilage tissues. (**K**) Comparison of the relative mRNA expression levels of *RANKL* in thoracic vertebrae. (**L**) Comparison of the relative mRNA expression levels of *RANKL* in leg cartilage tissue. (**M**) Comparison of the relative mRNA expression levels of *TNF-α* in thoracic vertebrae. (**N**) Comparison of the relative mRNA expression levels of *TNF-α* in leg cartilage tissue. * indicates significant difference (*p* < 0.05).

**Table 1 genes-13-01865-t001:** Diet composition and nutrition level.

Compositions of Diets %	Chick Feed	Flocks of Adult Feed
Corn	64.70	60.90
Soy protein	30.2	25.1
Wheat bran	0.00	10.00
Soybean oil	1.10	0.00
Calcium hydrogen phosphate	1.50	1.50
Stone meal	0.70	0.60
Middling flour	0.41	0.46
Met	0.08	0.07
Salt	0.35	0.35
Minerals and vitamins	1.00	1.00
Total	100	100
Nutrients levels		
Metabolism energy (Kcal·kg^−1^)	2900	2860
Crude protein	20.00	19.50
Calcium	0.90	0.80
Phosphorus	0.37	0.37
Lys	0.98	0.75
Met	0.39	0.32

Note: Main composition of premix (converted to ration per kg): VA 15,000 U, VD 33,300 U, VE 62.5 mg, VK 3.6 mg, VB_l_ 3 mg, VB_2_ 9 mg, VB_6_ 6 mg, VB_12_ 0.03 mg, Nicotinamide 60 mg, D-pantothenic acid 18 mg, Folic acid 1.5 mg, Biotin 0.36 mg, Choline chloride 600 mg, Fe 80 mg, Cu 12 mg, Zn 75 mg, Mn 60 mg, I 0.35 mg, Se 0.15 mg, and antibacterial growth promoters, antioxidants.

**Table 2 genes-13-01865-t002:** Vaccination protocol.

Age (Day)	Vaccine	Way
1	Marek’s disease	Subcutaneous injection
3	Newcastle	Oral vaccination
12	Gumboro	Subcutaneous injection
20	Newcastle	Oral vaccination
42	Fowl cholera	Subcutaneous injection

**Table 3 genes-13-01865-t003:** Dosage of medicines.

Group	Medicines	Dosage
NC	NA	NA
CHF-H	CHF	0.6%
CHF-M		0.4%
CHF-L		0.2%
BPs-H	BPS	0.6%
BPs-M		0.4%
BPs-L		0.2%

Note: NC: the experimental chickens were fed with normal diet; CHF-H: normal diet supplemented with 0.6% Chinese herbal formula was fed to experimental chickens; CHF-M: normal diet supplemented with 0.4% Chinese herbal formula was fed to experimental chickens; CHF-L: normal diet supplemented with 0.2% Chinese herbal formula was fed to experimental chickens; BPs-H: normal diet supplemented with 0.6% bisphosphonate was fed to experimental chickens. BPs-M: normal diet supplemented with 0.4% bisphosphonate was fed to experimental chickens. BPs-L: normal diet supplemented with 0.2% bisphosphonate was fed to experimental chickens.

**Table 4 genes-13-01865-t004:** Comparison of live weight and survival rate at 30 and 60 days old.

Item	NC	CHF-H	CHF-M	CHF-L	BPs-H	BPs-M	BPs-L
30d	941.91 ^abc^ ±70.02	1012.42 ^ab^ ±140.81	1058.02 ^a^ ±122.14	982.25 ^abc^ ±84.91	936.82 ^abc^ ±9.15	895.55 ^bc^ ±60.81	852.91 ^c^ ±14.79
60d	2603.56 ^a^ ±105.94	2620.85 ^a^ ±282.32	2612.83 ^a^ ±180.23	2612.14 ^a^ ±178.73	2358.85 ^b^ ±97.68	2364.74 ^b^ ±113.95	2251.21 ^b^ ±247.58
Survival rate	93.50%	98.00%	97.40%	96.00%	86.40%	88.00%	89.4%

Note: No same letter in the same row of shoulder marks indicates significant difference (*p* < 0.05).

## Data Availability

The data presented in this study are available on request from the corresponding author. The data are not publicly available due to privacy.

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
