# Peer review of "Influence of Chinese Herbal Formula on Bone Characteristics of Cobb Broiler Chickens"

_genes, 2022, doi:10.3390/genes13101865_

Round 1

Reviewer 1 Report

Manuscript Genes-1961569 entitled “Research on nutritional regulation technology of Chinese herbal formula to improve bone characteristics of Cobb broiler chicken. Please notice the following:

General view: The manuscript discussed a brilliant idea to improve poultry industry production but was expressed in weak language and grammar. The manuscript requires extensive proofreading to provide more meaningful sentences. 

The manuscript could be accepted for publication after minor revision.

Title: Clear to some extent but preferred to be modified into “Influence of Chinese Herbal Formula on Bone Characteristics of Cobb Broiler Chickens”.

Abstract: Please notice the following:

1.      L18: the expression “Cobb broilers were selected as experimental object” has to be replaced with “560-one-day-old Cobb broiler chickens were experimented for the influence of Chinese herbal formula (CHF) and Bisphosphonates (BPs).

2.      L19: remove the repeated “and”.

3. L23: please express these words “CHF group>normal control(NC) group>BPs group” in other meaningful words.

4.      L23: replace “could improve” with “improved”.

5.  L25-33: rephrase and divide into shorter sentences of more powerful meaning.

6.      The conclusion of the abstract section requires rephrasing.

Introduction: Properly displayed. into three paragraphs only i.e., 1. Introduction 2. Significance of the study, and 3. Aim of the study. Please notice the following:

1.      L67-81: Please rephrase to a better meaning for illustrating the facts of the herbs.

2.      L88-89: “pollution-free” was repeated twice, please delete it.

The aim: Clear and informative.

Materials and Methods: Please notice the following:

1.      L107-108: remove “of the floor area”.

2.   Please illustrate details on the microclimatic conditions of the broiler chickens such as temperature, relative humidity, lighting regimen, heating system, ventilation system, litter management, fly-prof, rodent-prof, and cleaning and disinfection program.

3.      Add a reference for the extraction of the Chinese herbal formula.

4.   The sampling of 30 out of 80 broiler chickens for measurement is non-indicative.

5.   The authors did not list the method used for the hygienic disposal of carcasses after sampling.

6.      Please provide the statistical model used in the SPSS analysis.

Results: Novel, clear, and informative. Please notice the following:

1.   Consider listing the p-value in this section after the “significant and highly significant” expressions in the results section.

2.     Do not list the p-value in case of non-significant results in the results section.

3.   The expression “CHF-H group>CHF-L group>BPs-H group” was listed in more than one situation and more than one shape, so please modify it in more meaningful words.

Discussion: Clear to some extent, informative, and contribute to knowledge with a moderate level of speculations and a good level of comparison. Many sentences have to be rephrased to provide a more powerful meaning.

Conclusion: Clear and informative.

Authors’ contributions: Clear and informative.

Funding: Clear and informative.

Acknowledgment: NA.

References: Excellent as only 81.25% (39 out of 48) were published in the past five years.

Tables: Well organized and presented.

Figures: Well organized and presented.

Reviewer 2 Report

Authors demonstrated the improvement in bone characteristics of Cobb broiler chicken by using Chinese herbal formula (CHF) which is comparable to that of Bisphosphonates (BPs).

The manuscript was well-written, and application of materials and methods is appropriate.

This reviewer noted following minor points:

1.       Meaning of CHF-H, -M, -L and BPs-H, -M, -L should be mentioned in the text or described as footnote of Table 3. How did author judge high, medium, low for CHF and BPs? It should be described in the materials and methods section.

2.       In Table 3, instead of “-----“, NA (not applicable) should be used.

3.       Line 176, “All the primer sequences (Table 4)”, this sentence has no meaning and rephrase it.

4.       Table 4 should be moved to supplementary material. Are these primers (table 4) newly designed in this study or previously reported primers?
